# Aligning Pretraining for Detection via Object-Level Contrastive Learning

**Fangyun Wei**∗ **Yue Gao**∗ **Zhirong Wu** **Han Hu** **Stephen Lin**

Microsoft Research Asia
{fawe, yuegao, wuzhiron, hanhu, stevelin}@microsoft.com

## Abstract

Image-level contrastive representation learning has proven to be highly effective as a generic model for transfer learning. Such generality for transfer learning, however, sacrifices specificity if we are interested in a certain downstream task. We argue that this could be sub-optimal and thus advocate a design principle which encourages alignment between the self-supervised pretext task and the downstream task. In this paper, we follow this principle with a pretraining method specifically designed for the task of object detection. We attain alignment in the following three aspects: 1) object-level representations are introduced via selective search bounding boxes as object proposals; 2) the pretraining network architecture incorporates the same dedicated modules used in the detection pipeline (e.g. FPN); 3) the pretraining is equipped with object detection properties such as object-level translation invariance and scale invariance. Our method, called Selective Object COntrastive learning (SoCo), achieves state-of-the-art results for transfer performance on COCO detection using a Mask R-CNN framework. Code is available at https://github.com/hologerry/SoCo.

## 1 Introduction

Pretraining and finetuning has been the dominant paradigm of training deep neural networks in computer vision. Downstream tasks usually leverage pretrained weights learned on large labeled datasets such as ImageNet [1] for initialization. As a result, supervised ImageNet pretraining has been prevalent throughout the field. Recently, self-supervised pretraining [2, 3, 4, 5, 6, 7, 8, 9] has achieved considerable progress and alleviated the dependency on labeled data. These methods aim to learn generic visual representations for various downstream tasks by means of image-level pretext tasks, such as instance discrimination. Some recent works [10, 11, 12, 13] observe that the image-level representations are sub-optimal for dense prediction tasks such as object detection and semantic segmentation. A potential reason is that image-level pretraining may overfit to holistic representations and fail to learn properties that are important outside of image classification.

The goal of this work is to develop self-supervised pretraining that is aligned to object detection. In object detection, bounding boxes are widely adopted as the representation for objects. Translation and scale invariance for object detection are reflected by the location and size of the bounding boxes. An obvious representation gap exists between image-level pretraining and the object-level bounding boxes of object detection.

Motivated by this, we present an object-level self-supervised pretraining framework, called **S**elective **O**bject **CO**ntrastive learning (SoCo), specifically for the downstream task of object detection. To introduce object-level representations into pretraining, SoCo utilizes off-the-shelf selective search [14] to generate object proposals. Different from prior image-level contrastive learning methods which

---

∗Equal contribution.

35th Conference on Neural Information Processing Systems (NeurIPS 2021).

treat the whole image as an instance, SoCo treats each object proposal in the image as an independent instance. This enables us to design a new pretext task for learning object-level visual representations with properties that are compatible with object detection. Specifically, SoCo constructs object-level views where the scales and locations of the same object instance are augmented. Contrastive learning follows to maximize the similarity of the object across augmented views.

The introduction of the object-level representation also allows us to further bridge the gap in network architecture between pretraining and finetuning. Object detection often involves dedicated modules, e.g., feature pyramid network (FPN) [15], and special-purpose sub-networks, e.g., R-CNN head [16, 17]. In contrast to image-level contrastive learning methods where only feature backbones are pretrained and transferred, SoCo performs pretraining over all the network modules used in detectors. As a result, all layers of the detectors can be well-initialized.

Experimentally, the proposed SoCo achieves state-of-the-art transfer performance from ImageNet to COCO. Concretely, by using Mask R-CNN with an R50-FPN backbone, it obtains 43.2 $AP^{bb}$ / 38.4 $AP^{mk}$ on COCO with a $1\times$ schedule, which are +4.3 $AP^{bb}$ / +3.0 $AP^{mk}$ better than the supervised pretraining baseline, and 44.3 $AP^{bb}$ / 39.6 $AP^{mk}$ on COCO with a $2\times$ schedule, which are +3.0 $AP^{bb}$ / +2.3 $AP^{mk}$ better than the supervised pretraining baseline. When transferred to Mask R-CNN with an R50-C4 backbone, it achieves 40.9 $AP^{bb}$ / 35.3 $AP^{mk}$ and 42.0 $AP^{bb}$ / 36.3 $AP^{mk}$ on the $1\times$ and $2\times$ schedule, respectively.

## 2   Related Work

Unsupervised feature learning has a long history in training deep neural networks. Auto-encoders [18] and Deep Boltzmann Machines [19] learn layers of representations by reconstructing image pixels. Unsupervised pretraining is often used as a method for initialization, which is followed by supervised training on the same dataset such as handwritten digits for classification.

Recent advances in unsupervised learning, especially self-supervised learning, tend to formulate the problem in a transfer learning scenario, where pretraining and finetuning are trained on different datasets with different purposes. Pretext tasks such as colorization [20], context prediction [21], inpainting [22] and rotation prediction [23] force the network to learn semantic information in order to solve the pretext tasks. Contrastive methods based on the instance discrimination pretext task [2] learn to map augmented views of the same instance to similar embeddings. Such approaches [24, 4, 5, 9, 25, 26] have shown strong transfer ability for a number of downstream tasks, sometimes even outperforming the supervised counterpart by large margins [4].

With new technical innovations such as learning clustering assignments [7], large-scale contrastive learning has been successfully applied on non-curated data sets [27, 28]. While the linear evaluation result on ImageNet classification has improved significantly [3], the progress on transfer performance for dense prediction tasks has been limited. Due to this, a growing number of works investigate pretraining specifically for object detection and semantic segmentation. The idea is to shift image-level representations to pixel-level or region-level representations. VADer [29], PixPro [10] and DenseCL [11] propose to learn pixel-level representations by matching point features of the same physical location under different views. InsLoc [12] learns to match region-level features from composited imagery. DetCon [13] additionally uses the bottom-up segmentation of MCG [30] for supervising intra-segment pixel variations. UP-DETR [31] proposes a pretext task named random query patch detection to pretrain DETR detector. Self-EMD [32] explores self-supervised representation learning for object detection without using Imagenet images.

Our work is also motivated and inspired by object detection methods which advocate architectures and training schemes invariant to translation and scale transformations [15, 33]. We find that incorporating them in the pretraining stage also benefits object detection, which has largely been overlooked in previous self-supervised learning works.

## 3   Method

We introduce a new contrastive learning method which maximizes the similarity of object-level features representing different augmentations of the same object. Meanwhile, we introduce architectural alignment and important properties of object detection into pretraining when designing the object-level pretext task. We use an example where transfer learning is performed on Mask-RCNN with a R50-FPN backbone to illustrate the core design principles. To further demonstrate the extensibility

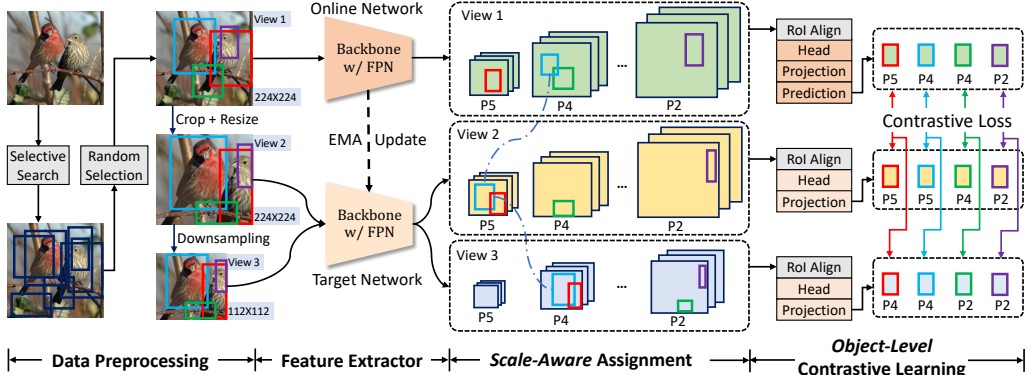

Figure 1: Overview of SoCo. SoCo utilizes selective search to generate a set of object proposals for each raw image. $K$ proposals are randomly selected in each training step. We construct three views $\{V_1, V_2, V_3\}$ where the scales and locations of the same object are different. We adopt a backbone with FPN to encode image-level features and RoIAlign to extract object-level features. Object proposals are assigned to different pyramid levels according to their image areas. Contrastive learning is performed at the object level to learn translation-invariant and scale-invariant representations. The target network is updated by an exponential moving average of the online network.

and flexibility of our method, we also apply SoCo on a R50-C4 structure. In this section, we first present an overview of the proposed SoCo in Section 3.1. Then we describe the process of object proposal generation and view construction in Section 3.2. Finally, the object-level contrastive learning and our design principles are introduced in Section 3.3.

## 3.1 Overview

Figure 1 displays an overview of SoCo. SoCo aims to align pretraining to object detection in two aspects: 1) network architecture alignment between pretraining and object detection; 2) introducing central properties of detection. Concretely, besides pretraining a backbone as done in existing self-supervised contrastive learning methods, SoCo also pretrains all the network modules used in an object detector, such as FPN and the head in the Mask R-CNN framework. As a result, all layers of the detector can be well-initialized. Furthermore, SoCo strives to learn object-level representations which are not only meaningful, but also invariant to translation and scale. To achieve this, it encourages diversity of scales and locations of objects by constructing multiple augmented views and applying a scale-aware assignment strategy for different levels of a feature pyramid. Finally, object-level contrastive learning is applied to maximize the feature similarity of the same object across augmented views.

## 3.2 Data Preprocessing

**Object Proposal Generation.** Inspired by R-CNN [34] and Fast R-CNN [35], we use selective search [14], an unsupervised object proposal generation algorithm which takes into account color similarity, texture similarity, size of region and fit between regions, to generate a set of object proposals for each of the raw images. We represent each object proposal as a bounding box $b = \{x, y, w, h\}$, where $(x, y)$ denotes the coordinates of the bounding box center, and $w$ and $h$ are the corresponding width and height, respectively. We keep only the proposals that satisfy the following requirements: 1) $1/3 \leq w/h \leq 3$; 2) $0.3 \leq \sqrt{wh}/\sqrt{WH} \leq 0.8$, where $W$ and $H$ denote width and height of the input image. The object proposal generation step is performed offline. In each training iteration, we randomly select $K$ proposals for each input image.

**View Construction.** Three views, namely $V_1$, $V_2$ and $V_3$, are used in SoCo. The input image is resized to $224 \times 224$ to obtain $V_1$. Then we apply a random crop with a scale range of [0.5, 1.0] on $V_1$ in generating $V_2$. $V_2$ is then resized to the same size as $V_1$ and object proposals outside of $V_2$ are dropped. Next, we downsample $V_2$ to a fixed size (e.g. $112 \times 112$) to produce $V_3$. In all of these cases, the bounding boxes transformed according to the cropping and resizing of the RGB images

(see Figure 1, *Data Preprocessing*). Finally, each view is randomly and independently augmented. We adopt the augmentation pipeline of BYOL [3] but discard the random crop augmentation since spatial transformation is already applied on all three views. Notice that the scale and location of the same object proposal are different across the augmented views, which enables the model to learn translation-invariant and scale-invariant object-level representations.

**Box Jitter.** To further encourage variance of scales and locations of object proposals across views, we adopt a box jitter strategy on the generated proposals as an object-level data augmentation. Specifically, given an object proposal $b = \{x, y, w, h\}$, we randomly generate a jittered box $\hat{b} = \{\hat{x}, \hat{y}, \hat{w}, \hat{h}\}$ as follows: 1) $\hat{x} = x + r \cdot w$; 2) $\hat{y} = y + r \cdot h$; 3) $\hat{w} = w + r \cdot w$; 4) $\hat{h} = h + r \cdot h$, where $r \in [-0.1, 0.1]$. The box jitter is randomly applied on each proposal with a probability of $0.5$.

### 3.3 Object-Level Contrastive Learning

The goal of SoCo is to align pretraining to object detection. Here, we use the representative framework Mask R-CNN [17] with feature pyramid network (FPN) [15] to demonstrate our key design principles. The alignment mainly involves aligning the pretraining architecture with that of object detection and integrating important object detection properties such as object-level translation invariance and scale invariance into the pretraining.

**Aligning Pretraining Architecture to Object Detection.** Following Mask R-CNN, we use a backbone with FPN as the image-level feature extractor $f^I$. We denote the output of FPN as $\{P_2, P_3, P_4, P_5\}$ with a stride of $\{4, 8, 16, 32\}$. Here, we do not use $P_6$ due to its low resolution. With the bounding box representation $b$, RoIAlign [17] is applied to extract the foreground feature from the corresponding scale level. For further architectural alignment, we additionally introduce an R-CNN head $f^H$ into pretraining. The object-level feature representation $h$ of bounding box $b$ is extracted from an image view $V$ as:

$$h = f^H(\text{RoIAlign}(f^I(V), b)). \tag{1}$$

SoCo uses two neural networks to learn, namely an online network and a target network. The online network and the target network share the same architecture but with different sets of weights. Concretely, the target network weights $f_\xi^I$, $f_\xi^H$ are the exponential moving average (EMA) with the momentum coefficient $\tau$ of the online parameters $f_\theta^I$ and $f_\theta^H$. Denote the set of object proposals $\{b_i\}$ considered in the image. Let $h_i$ be the object-level representation of proposal $b_i$ in view $V_1$, and $h_i'$, $h_i''$ be the representation of $b_i$ in view $V_2$, $V_3$. They are extracted using the online network and the target network respectively,

$$h_i = f_\theta^H(\text{RoIAlign}(f_\theta^I(V_1), b_i)), \tag{2}$$

$$h_i' = f_\xi^H(\text{RoIAlign}(f_\xi^I(V_2), b_i)), \quad h_i'' = f_\xi^H(\text{RoIAlign}(f_\xi^I(V_3), b_i)). \tag{3}$$

We follow BYOL [3] for learning contrastive representations. The online network is appended with a projector $g_\theta$, and a predictor $q_\theta$ for obtaining latent embeddings. Both $g_\theta$ and $q_\theta$ are two-layer MLPs. The target network is only appended with the projector $g_\xi$ for avoiding trivial solutions. We use $v_i$, $v_i'$ and $v_i''$ to denote the latent embeddings of object-level representations $\{h_i, h_i', h_i''\}$, respectively:

$$v_i = q_\theta(g_\theta(h_i)), \quad v_i' = g_\xi(h_i'), \quad v_i'' = g_\xi(h_i''). \tag{4}$$

The contrastive loss for the $i$-th object proposal is defined as:

$$\mathcal{L}_i = -2 \cdot \frac{\langle v_i, v_i' \rangle}{\|v_i\|_2 \cdot \|v_i'\|_2} - 2 \cdot \frac{\langle v_i, v_i'' \rangle}{\|v_i\|_2 \cdot \|v_i''\|_2}. \tag{5}$$

Then we can formulate the overall loss function for each image as:

$$\mathcal{L} = \frac{1}{K} \sum_{i=1}^{K} \mathcal{L}_i, \tag{6}$$

where $K$ is the number of object proposals. We symmetrize the loss $\mathcal{L}$ in Eq. 6 by separately feeding $V_1$ to the target network and $\{V_2, V_3\}$ to the online network to compute $\widetilde{\mathcal{L}}$. At each training iteration, we perform a stochastic optimization step to minimize $\mathcal{L}^{\text{SoCo}} = \mathcal{L} + \widetilde{\mathcal{L}}$.

**Scale-Aware Assignment.** Mask R-CNN with FPN uses the IoU between anchors and ground-truth boxes to determine positive samples. It defines anchors to have areas of $\{32^2, 64^2, 128^2, 256^2\}$ pixels on $\{P_2, P_3, P_4, P_5\}$, respectively, which means ground-truth boxes of scale within a range are assigned to a specific pyramid level. Inspired by this, we propose a scale-aware assignment strategy, which greatly encourages the pretraining model to learn object-level scale-invariant representations. Concretely, we assign object proposals of area within a range of $\{0 - 48^2, 49^2 - 96^2, 97^2 - 192^2, 193^2 - 224^2\}$ pixels to $\{P_2, P_3, P_4, P_5\}$, respectively. Notice that the maximum proposal size is $224 \times 224$ since all of the views are resized to a fixed resolution smaller than 224. The advantage is that the same object proposals at different scales are encouraged to learn consistent representations through contrastive learning. As a result, SoCo learns object-level scale-invariant visual representations, which is important for object detection.

**Introducing Properties of Detection to Pretraining.** Here, we discuss how SoCo promotes important properties of object detection in the pretraining. Object detection uses tight bounding boxes to represent objects. To introduce object-level representations, SoCo generates object proposals by selective search. Translation invariance and scale invariance at the object level are regarded as the most important properties for object detection, i.e., feature representations of objects belonging to same category should be insensitive to scale and location. Recall that $V_2$ is a randomly cropped patch of $V_1$. Random cropping introduces box shift and thus contrastive learning between $V_1$ and $V_2$ encourages the pretraining model to learn location-invariant representations. $V_3$ is generated by downsampling $V_2$, which results in a scale augmentation of object proposals. With our scale-aware assignment strategy, the contrastive loss between $V_1$ and $V_3$ guides the pretraining towards learning scale-invariant visual representations.

**Extending to Other Object Detectors.** SoCo can be easily extended to align to other detectors besides Mask R-CNN with FPN. Here, we apply SoCo to Mask R-CNN with a C4 structure, which is a popular non-FPN detection framework. The modification is three-fold: 1) for all object proposals, RoIAlign is performed on $C_4$; 2) the R-CNN head is replaced by the entire 5-th residual block; 3) view $V_3$ is discarded and the remaining $V_1$ and $V_2$ are kept for object-level contrastive learning. Experiments and comparisons with state-of-the-art methods in Section 4.3 demonstrate the extensibility and flexibility of SoCo.

# 4 Experiments

## 4.1 Pretraining Settings

**Architecture.** Through the introduction of object proposals, the architectural discrepancy is reduced between pretraining and downstream detection finetuning. Mask R-CNN [17] is a commonly adopted framework to evaluate transfer performance. To demonstrate the extensibility and flexibility of SoCo, we provide details of SoCo alignment for the detection architectures R50-FPN and R50-C4. *SoCo-R50-FPN*: ResNet-50 [36] with FPN [15] is used as the image-level feature encoder. RoIAlign [17] is then used to extract RoI features on feature maps $\{P_2, P_3, P_4, P_5\}$ with a stride of $\{4, 8, 16, 32\}$. According to the image areas of object proposals, each RoI feature is then transformed to an object-level representation by the head network as in Mask R-CNN. *SoCo-R50-C4*: on the standard ResNet-50 architecture, we insert the RoI operation on the output of the 4-th residual block. The entire 5-th residual block is treated as the head network to encode object-level features. Both the projection network and prediction network are 2-layer MLPs which consist of a linear layer with output size 4096 followed by batch normalization [37], rectified linear units (ReLU) [38], and a final linear layer with output dimension 256.

**Dataset.** We adopt the widely used ImageNet [1] which consists of ~1.28 million images for self-supervised pretraining.

**Data Augmentation.** Once all views are constructed, we employ the data augmentation pipeline of BYOL [3]. Specifically, we apply random horizontal flip, color distortion, Gaussian blur, grayscaling, and the solarization operation. We remove the random crop augmentation since spatial transformations have already been applied on all views.

**Optimization.** We use a 100-epoch training schedule in all the ablation studies and report the results of 100-epochs and 400-epochs in the comparisons with state-of-the-art methods. We use the LARS optimizer [39] with a cosine decay learning rate schedule [40] and a warm-up period of 10 epochs.

Table 1: Comparison with state-of-the-art methods on **COCO** by using Mask R-CNN with **R50-FPN**.

| Methods | Epoch | 1× Schedule | | | | | | 2× Schedule | | | | | |
|---|---|---|---|---|---|---|---|---|---|---|---|---|---|
| | | $AP^{bb}$ | $AP^{bb}_{50}$ | $AP^{bb}_{75}$ | $AP^{mk}$ | $AP^{mk}_{50}$ | $AP^{mk}_{75}$ | $AP^{bb}$ | $AP^{bb}_{50}$ | $AP^{bb}_{75}$ | $AP^{mk}$ | $AP^{mk}_{50}$ | $AP^{mk}_{75}$ |
| Scratch | - | 31.0 | 49.5 | 33.2 | 28.5 | 46.8 | 30.4 | 38.4 | 57.5 | 42.0 | 34.7 | 54.8 | 37.2 |
| Supervised | 90 | 38.9 | 59.6 | 42.7 | 35.4 | 56.5 | 38.1 | 41.3 | 61.3 | 45.0 | 37.3 | 58.3 | 40.3 |
| MoCo [4] | 200 | 38.5 | 58.9 | 42.0 | 35.1 | 55.9 | 37.7 | 40.8 | 61.6 | 44.7 | 36.9 | 58.4 | 39.7 |
| MoCo v2 [5] | 200 | 40.4 | 60.2 | 44.2 | 36.4 | 57.2 | 38.9 | 41.7 | 61.6 | 45.6 | 37.6 | 58.7 | 40.5 |
| InfoMin [6] | 200 | 40.6 | 60.6 | 44.6 | 36.7 | 57.7 | 39.4 | 42.5 | 62.7 | 46.8 | 38.4 | 59.7 | 41.4 |
| BYOL [3] | 300 | 40.4 | 61.6 | 44.1 | 37.2 | 58.8 | 39.8 | 42.3 | 62.6 | 46.2 | 38.3 | 59.6 | 41.1 |
| SwAV [7] | 400 | - | - | - | - | - | - | 42.3 | 62.8 | 46.3 | 38.2 | 60.0 | 41.0 |
| ReSim-FPN$^T$ [45] | 200 | 39.8 | 60.2 | 43.5 | 36.0 | 57.1 | 38.6 | 41.4 | 61.9 | 45.4 | 37.5 | 59.1 | 40.3 |
| PixPro [10] | 400 | 41.4 | 61.6 | 45.4 | - | - | - | - | - | - | - | - | - |
| InsLoc [12] | 400 | 42.0 | 62.3 | 45.8 | 37.6 | 59.0 | 40.5 | 43.3 | 63.6 | 47.3 | 38.8 | 60.9 | 41.7 |
| DenseCL [11] | 200 | 40.3 | 59.9 | 44.3 | 36.4 | 57.0 | 39.2 | 41.2 | 61.9 | 45.1 | 37.3 | 58.9 | 40.1 |
| DetCon$_S$ [13] | 1000 | 41.8 | - | - | 37.4 | - | - | 42.9 | - | - | 38.1 | - | - |
| DetCon$_B$ [13] | 1000 | 42.7 | - | - | 38.2 | - | - | 43.4 | - | - | 38.7 | - | - |
| SoCo | 100 | 42.3 | 62.5 | 46.5 | 37.6 | 59.1 | 40.5 | 43.2 | 63.3 | 47.3 | 38.8 | 60.6 | 41.9 |
| SoCo | 400 | 43.0 | 63.3 | 47.1 | 38.2 | 60.2 | 41.0 | 44.0 | 64.0 | 48.4 | 39.0 | 61.3 | 41.7 |
| **SoCo*** | 400 | **43.2** | **63.5** | **47.4** | **38.4** | **60.2** | **41.4** | **44.3** | **64.6** | **48.9** | **39.6** | **61.8** | **42.5** |

The base learning rate $lr_{base}$ is set to 1.0 and is scaled linearly [41] with the batch size ($lr = lr_{base} \times$ BatchSize/256). The weight decay is set to $1.0 \times e^{-5}$. The total batch size is set to 2048 over 16 Nvidia V100 GPUs. For the update of the target network, following [3], the momentum coefficient $\tau$ starts from 0.99 and is increased to 1 during training. Synchronized batch normalization is enabled.

## 4.2 Transfer Learning Settings

COCO [42] and Pascal VOC [43] datasets are used for transfer learning. `Detectron2` [44] is used as the code base.

**COCO Object Detection and Instance Segmentation.** We use the COCO `train2017` set which contains ∼118k images with bounding box and instance segmentation annotations in 80 object categories. Transfer performance is evaluated on the COCO `val2017` set. We adopt the Mask R-CNN detector [17] with the R50-FPN and the R50-C4 backbones. We report $AP^{bb}$, $AP^{bb}_{50}$ and $AP^{bb}_{75}$ for object detection, and $AP^{mk}$, $AP^{mk}_{50}$ and $AP^{mk}_{75}$ for instance segmentation. We report the results under the COCO 1× and 2× schedules in comparisons with state-of-the-art methods. All ablation studies are conducted on the COCO 1× schedule.

**Pascal VOC Object Detection.** We use the Pascal VOC `trainval07+12` set which contains ∼16.5k images with bounding box annotations in 20 object categories as the training set. Transfer performance is evaluated on the Pascal VOC `test2007` set. We report the results of $AP^{bb}$, $AP^{bb}_{50}$ and $AP^{bb}_{75}$ for object detection.

**Optimization.** All the pretrained weights except for projection and prediction are loaded into the object detection network for transfer. Following [5], synchronized batch normalization is used across all layers including the newly initialized batch normalization layers in finetuning. For both the COCO and Pascal VOC datasets, we finetune with stochastic gradient descent and a batch size of 16 split across 8 GPUs. For COCO transfer, we use a weight decay of $2.5 \times e^{-5}$, and a base learning rate of 0.02 that increases linearly for the first 1000 iterations and drops twice by a factor of 10, after $\frac{2}{3}$ and $\frac{8}{9}$ of the total training time. For Pascal VOC transfer, the weight decay is $1.0 \times e^{-4}$, and the base learning rate is set to 0.02 and divided by 10 at $\frac{3}{4}$ and $\frac{11}{12}$ of the total training time.

## 4.3 Comparison with State-of-the-Art Methods

**Mask R-CNN with R50-FPN on COCO.** Table 1 shows the transfer results for Mask R-CNN with R50-FPN backbone. We compare SoCo with the state-of-the-art unsupervised pretraining methods on the COCO 1× and 2× schedules. We report our results under 100 epochs and 400 epochs of pretraining. On the 100-epoch training schedule, SoCo achieves 42.3 $AP^{bb}$ / 37.6 $AP^{mk}$ and 43.2 $AP^{bb}$ / 38.8 $AP^{mk}$ on the 1× and 2× schedules, respectively. Pretrained for 400 epochs, SoCo outperforms all previous pretraining methods designed for either image classification or object detection, achieving 43.0 $AP^{bb}$ / 38.2 $AP^{mk}$ for the 1× schedule and 44.0 $AP^{bb}$ / 39.0 $AP^{mk}$ for the 2×

Table 2: Comparison with state-of-the-art methods on **COCO** using Mask R-CNN with **R50-C4**.

| Methods | Epoch | 1× Schedule | | | | | | 2× Schedule | | | | | |
|---|---|---|---|---|---|---|---|---|---|---|---|---|---|
| | | $AP^{bb}$ | $AP^{bb}_{50}$ | $AP^{bb}_{75}$ | $AP^{mk}$ | $AP^{mk}_{50}$ | $AP^{mk}_{75}$ | $AP^{bb}$ | $AP^{bb}_{50}$ | $AP^{bb}_{75}$ | $AP^{mk}$ | $AP^{mk}_{50}$ | $AP^{mk}_{75}$ |
| Scratch | - | 26.4 | 44.0 | 27.8 | 29.3 | 46.9 | 30.8 | 35.6 | 54.6 | 38.2 | 31.4 | 51.5 | 33.5 |
| Supervised | 90 | 38.2 | 58.2 | 41.2 | 33.3 | 54.7 | 35.2 | 40.0 | 59.9 | 43.1 | 34.7 | 56.5 | 36.9 |
| MoCo [4] | 200 | 38.5 | 58.3 | 41.6 | 33.6 | 54.8 | 35.6 | 40.7 | 60.5 | 44.1 | 35.4 | 57.3 | 37.6 |
| SimCLR [9] | 200 | - | - | - | - | - | - | 39.6 | 59.1 | 42.9 | 34.6 | 55.9 | 37.1 |
| MoCo v2 [5] | 800 | 39.3 | 58.9 | 42.5 | 34.3 | 55.7 | 36.5 | 41.2 | 60.9 | 44.6 | 35.8 | 57.7 | 38.2 |
| InfoMin [6] | 200 | 39.0 | 58.5 | 42.0 | 34.1 | 55.2 | 36.3 | 41.3 | 61.2 | 45.0 | 36.0 | 57.9 | 38.3 |
| BYOL [3] | 300 | - | - | - | - | - | - | 40.3 | 60.5 | 43.9 | 35.1 | 56.8 | 37.3 |
| SwAV [7] | 400 | - | - | - | - | - | - | 39.6 | 60.1 | 42.9 | 34.7 | 56.6 | 36.6 |
| SimSiam [8] | 200 | 39.2 | 59.3 | 42.1 | 34.4 | 56.0 | 36.7 | - | - | - | - | - | - |
| PixPro [10] | 400 | 40.5 | 59.8 | 44.0 | - | - | - | - | - | - | - | - | - |
| InsLoc [12] | 400 | 39.8 | 59.6 | 42.9 | 34.7 | 56.3 | 36.9 | 41.8 | 61.6 | 45.4 | **36.3** | 58.2 | **38.8** |
| SoCo | 100 | 40.4 | 60.4 | 43.7 | 34.9 | 56.8 | 37.0 | 41.1 | 61.0 | 44.4 | 35.6 | 57.5 | 38.0 |
| **SoCo** | 400 | **40.9** | **60.9** | **44.3** | **35.3** | **57.5** | **37.3** | **42.0** | **61.8** | **45.6** | **36.3** | **58.5** | **38.8** |

Table 3: Comparison with state-of-the-art methods on **Pascal VOC**. Faster R-CNN with **R50-C4** is adopted. Table is split to two sub-tables.

(a) Sub-table 1.

| Methods | Epoch | $AP^{bb}$ | $AP^{bb}_{50}$ | $AP^{bb}_{75}$ |
|---|---|---|---|---|
| Scratch | - | 33.8 | 60.2 | 33.1 |
| Supervised | 90 | 53.5 | 81.3 | 58.8 |
| ReSim-C4 [45] | 200 | 58.7 | 83.1 | 66.3 |
| PixPro [10] | 400 | **60.2** | **83.8** | **67.7** |
| InsLoc [12] | 400 | 58.4 | 83.0 | 65.3 |
| DenseCL [11] | 200 | 58.7 | 82.8 | 65.2 |
| SoCo | 100 | 59.1 | 83.4 | 65.6 |
| SoCo | 400 | 59.7 | **83.8** | 66.8 |

(b) Sub-table 2.

| Methods | Epoch | $AP^{bb}$ | $AP^{bb}_{50}$ | $AP^{bb}_{75}$ |
|---|---|---|---|---|
| MoCo [4] | 200 | 55.9 | 81.5 | 62.6 |
| SimCLR [9] | 1000 | 56.3 | 81.9 | 62.5 |
| MoCo v2 [5] | 800 | 57.6 | 82.7 | 64.4 |
| InfoMin [6] | 200 | 57.6 | 82.7 | 64.6 |
| BYOL [3] | 300 | 51.9 | 81.0 | 56.5 |
| SwAV [7] | 400 | 45.1 | 77.4 | 46.5 |
| SimSiam [8] | 200 | 57.0 | 82.4 | 63.7 |
| ReSim-FPN [45] | 200 | 59.2 | 82.9 | 65.9 |

schedule. Moreover, we propose an enhanced version named *SoCo\**, which constructs an additional view $V_4$ for pretraining. Similar to the construction step of $V_2$, $V_4$ is a randomly cropped patch of $V_1$. We resize $V_4$ to $192 \times 192$ and perform the same forward process as for $V_2$ and $V_3$. Compared with SoCo, SoCo* further boosts the performance and obtains an improvement of +0.2 $AP^{bb}$ / +0.2 $AP^{mk}$ on the 1× schedule, achieving 43.2 $AP^{bb}$ / 38.4 $AP^{mk}$, and +0.3 $AP^{bb}$ / +0.6 $AP^{mk}$ on the 2× schedule, achieving 44.3 $AP^{bb}$ / 39.6 $AP^{mk}$, respectively.

**Mask R-CNN with R50-C4 on COCO.** To demonstrate the extensibility and flexibility of SoCo, we also evaluate transfer performance using Mask R-CNN with R50-C4 backbone on the COCO benchmark. We report the results of SoCo under 100 epochs and 400 epochs. Table 2 compares the proposed method to previous state-of-the-art methods. Without bells and whistles, SoCo obtains state-of-the-art performance, achieving 40.9 $AP^{bb}$ / 35.3 $AP^{mk}$ and 42.0 $AP^{bb}$ / 36.3 $AP^{mk}$ on the COCO 1× and 2× schedules, respectively.

**Faster R-CNN with R50-C4 on Pascal VOC.** We also evaluate the transfer ability of SoCo on the Pascal VOC benchmark. Following [5], we adopt Faster R-CNN with R50-C4 backbone for transfer learning. Table 3 shows the comparison. SoCo obtains an improvement of +6.2 $AP^{bb}$ against the supervised pretraining baseline, achieving 59.7 $AP^{bb}$ for VOC object detection.

## 4.4 Ablation Study

To further understand the advantages of SoCo, we conduct a series of ablation studies that examine the effectiveness of object-level contrastive learning, the effects of alignment between pretraining and detection, and different hyper-parameters. For all ablation studies, we use Mask R-CNN with R50-FPN backbone for transfer learning and adopt a 100-epoch SoCo pretraining schedule. Transfer performance is evaluated under the COCO 1× schedule. For the ablation of each hyper-parameter or component, we fix all other hyper-parameters to the following default settings: view $V_3$ with $112 \times 112$ resolution, batch size of 2048, momentum coefficient $\tau = 0.99$, proposal number $K = 4$,

Table 4: Ablation study on the effectiveness of aligning pretraining to object detection.

| Whole Image | Selective Search | FPN | Head | Scale-aware Assignment | Box Jitter | Multi View | $AP^{bb}$ | $AP^{mk}$ |
|---|---|---|---|---|---|---|---|---|
| ✓ | | | | | | | 38.1 | 34.4 |
| ✓ | ✓ | | | | | | 40.6 (+2.5) | 36.8 (+2.4) |
| ✓ | ✓ | ✓ | | | | | 40.2 (+2.1) | 36.2 (+1.8) |
| ✓ | ✓ | ✓ | ✓ | | | | 41.2 (+3.1) | 37.0 (+2.6) |
| ✓ | ✓ | ✓ | ✓ | ✓ | | | 41.6 (+3.5) | 37.3 (+2.9) |
| ✓ | ✓ | ✓ | ✓ | ✓ | ✓ | | 41.7 (+3.6) | 37.5 (+3.1) |
| ✓ | ✓ | ✓ | ✓ | ✓ | ✓ | ✓ | **42.3 (+4.2)** | **37.6 (+3.2)** |

Table 5: Ablation studies on hyper-parameters for the proposed SoCo method.

(a) Study on image size of view $V_3$.

| Image Size | $AP^{bb}$ | $AP^{mk}$ |
|---|---|---|
| 96 | 42.1 | 37.7 |
| **112** | **42.3** | 37.6 |
| 128 | 42.1 | 37.7 |
| 160 | 42.0 | 37.6 |
| 192 | 42.2 | **37.8** |

(b) Study on batch size.

| Batch Size | $AP^{bb}$ | $AP^{mk}$ |
|---|---|---|
| 512 | 41.7 | **37.6** |
| 1024 | 41.9 | **37.6** |
| **2048** | **42.3** | **37.6** |
| 4096 | 41.4 | 37.3 |

(c) Study on proposal generation and proposal number $K$.

| Selective Search | Random | $K$ | $AP^{bb}$ | $AP^{mk}$ |
|---|---|---|---|---|
| ✓ | | 1 | 41.6 | 37.3 |
| ✓ | | **4** | **42.3** | **37.6** |
| ✓ | | 8 | 41.6 | 37.4 |
| ✓ | | 16 | 41.2 | 37.0 |
| | ✓ | 1 | 41.4 | 36.9 |
| | ✓ | 4 | NaN | NaN |
| | ✓ | 8 | NaN | NaN |

(d) Study on momentum coefficient $\tau$.

| $\tau$ | $AP^{bb}$ | $AP^{mk}$ |
|---|---|---|
| 0.98 | 35.0 | 31.7 |
| **0.99** | **42.3** | **37.6** |
| 0.993 | 41.8 | **37.6** |

box jitter and scale-aware assignment are used, FPN and R-CNN head are pretrained and transferred, and selective search is used as the proposal generator.

**Effectiveness of Aligning Pretraining to Object Detection.** We ablate each component of SoCo step by step to demonstrate the the effectiveness of aligning pretraining to object detection. Table 4 reports the studies. The baseline treats the whole image as an instance, without considering any detection properties or architecture alignment. It obtains 38.1 $AP^{bb}$ and 34.4 $AP^{mk}$. Selective search introduces object-level representations. By leveraging generated object proposals and conducting contrastive learning at the object level, our method obtains an improvement of +2.5 $AP^{bb}$ and +2.4 $AP^{mk}$. Next, we further minimize the architectural discrepancy between pretraining and object detection pipeline by introducing FPN and an R-CNN head into the pretraining architecture. We observe that only introducing FPN into pretraining slightly hurt the performance. In contrast, including both FPN and the R-CNN head improves the transfer performance to 41.2 $AP^{bb}$ / 37.0 $AP^{mk}$, which verifies the effectiveness of architectural alignment between self-supervised pretraining and downstream tasks. On top of architectural alignment, we further leverage scale-aware assignment which encourages scale-invariant object-level visual representations, and an improvement of +3.5 $AP^{bb}$ / +2.9 $AP^{mk}$ against the baseline is found. The box jitter strategy slightly elevates performance. Finally, multiple views ($V_3$) further directs SoCo towards learning scale-invariant and translation-invariant representations, and our method achieves 42.3 $AP^{bb}$ and 37.6 $AP^{mk}$.

**Ablation Study on Hyper-Parameters.** Table 5 examines sensitivity to the hyper-parameters of SoCo.

Table 5(a) ablates the resolution of view $V_3$. SoCo is found to be insensitive to the image size of $V_3$. We use $112 \times 112$ as the default resolution due to its slightly better transfer performance and low training computation cost.

Table 6: Transfer Learning on **LVIS** dataset using Mask R-CNN with **R50-FPN**.

| Method | Epoch | $1\times$ Schedule | | | | | | $2\times$ Schedule | | | | | |
|---|---|---|---|---|---|---|---|---|---|---|---|---|---|
| | | $AP^{bb}$ | $AP^{bb}_{50}$ | $AP^{bb}_{75}$ | $AP^{mk}$ | $AP^{mk}_{50}$ | $AP^{mk}_{75}$ | $AP^{bb}$ | $AP^{bb}_{50}$ | $AP^{bb}_{75}$ | $AP^{mk}$ | $AP^{mk}_{50}$ | $AP^{mk}_{75}$ |
| Supervised | 90 | 20.4 | 32.9 | 21.7 | 19.4 | 30.6 | 20.5 | 23.4 | 36.9 | 24.9 | 22.3 | 34.7 | 23.5 |
| **SoCo*** | 400 | **26.3** | **41.2** | **27.8** | **25.0** | **38.5** | **26.8** | **28.3** | **43.5** | **30.7** | **26.9** | **41.1** | **28.7** |

Table 7: Transfer learning on **RetinaNet** and **FCOS**.

(a) Transfer to RetinaNet.

| Method | Epoch | $AP^{bb}$ | $AP^{bb}_{50}$ | $AP^{bb}_{75}$ |
|---|---|---|---|---|
| Supervised | 90 | 36.3 | 55.3 | 38.6 |
| **SoCo*** | 400 | **38.3** | **57.2** | **41.2** |

(b) Transfer to FCOS.

| Method | Epoch | $AP^{bb}$ | $AP^{bb}_{50}$ | $AP^{bb}_{75}$ |
|---|---|---|---|---|
| Supervised | 90 | 36.6 | 56.0 | 38.8 |
| **SoCo*** | 400 | **37.4** | **56.3** | **39.9** |

Table 5(b) ablates the training batch size of SoCo. It can be seen that larger or smaller batch sizes hurt the performance. We use a batch size of 2048 by default.

Table 5(c) ablates the effectiveness of object proposal generation strategies. To demonstrate the superiority of meaningful proposals generated by selective search, we randomly generate $K$ bounding boxes in each of the training images to replace the selective search proposals (namely *Random* in Table 5(c)). SoCo can still yield satisfactory results using a single random bounding box to learn object-level representations, which is not surprising, since the RoIAlign operation performed on the randomly generated bounding boxes also introduces object-level representation learning. This fact indirectly demonstrates that downstream object detectors can benefit from self-supervised pretraining that involves object-level representations. However, the result is still slightly worse than the case where only one object proposal generated by selective search is used in the pretraining. For the cases where we randomly generate $4$ and $8$ bounding boxes, the noisy and meaningless proposals cause the pretraining to diverge. From the table, we also find that applying more object proposals ($K = 8$ and $K = 16$) generated by selective search can be harmful. One potential reason is that most of the images in the ImageNet dataset only contain a few objects, so a larger $K$ may create duplicated and redundant proposals.

Table 5(d) ablates the momentum coefficient $\tau$ of the exponential moving average, and $\tau = 0.99$ yields the best performance.

## 4.5 More Experiments

**Transfer Learning on LVIS Dataset.** In addition to COCO and PASCAL VOC object detection, we also consider the challenging LVIS v1 dataset [46] to demonstrate the effectiveness and generality of our approach. The LVIS v1 detection benchmark contains 1203 categories in a long-tailed distribution with few training samples. It is considered to be more challenging than the COCO benchmark. We use Mask R-CNN with R50-FPN backbone and follow the standard LVIS $1\times$ and $2\times$ training schedules. We do not use any training or post-processing tricks. Table 6 shows the results. We can see that our method achieves +5.9 $AP^{bb}$/+5.6 $AP^{mk}$ and +4.9 $AP^{bb}$/+4.6 $AP^{mk}$ improvements under the LVIS $1\times$ and $2\times$ training schedules, respectively.

**Transfer Learning on Different Object Detectors.** In addition to two-stage detectors, we further conduct experiments on RetinaNet [47] and FCOS [48], which are representative works for single-stage detectors and anchor-free detectors. We transfer the weights of backbone and FPN layers learned by SoCo* to RetinaNet and FCOS architectures and follow the standard COCO $1\times$ schedule. We use MMDetection [49] as code base and default optimization configs are adopted. Table 7 shows the comparison.

**Evaluation on Mini COCO.** Transfer to the full COCO dataset may be of limited significance due to the extensive supervision available from its large-scale annotated training data. To further demonstrate the generalization ability of SoCo, we conduct an experiment on a mini version of the COCO dataset, named Mini COCO. Concretely, we randomly select 5% or 10% of the training data from COCO `train2017` to form Mini COCO benchmarks. Mask-RCNN with R50-FPN backbone is adopted to evaluate the transfer performance on the COCO $1\times$ schedule. COCO `val2017` is still used

Table 8: Results on **Mini COCO 1×** schedule. Mask R-CNN with **R50-FPN** backbone is adopted.

| Methods | Epoch | Mini COCO (5%) | | | | | | Mini COCO (10%) | | | | | |
|---|---|---|---|---|---|---|---|---|---|---|---|---|---|
| | | $AP^{bb}$ | $AP^{bb}_{50}$ | $AP^{bb}_{75}$ | $AP^{mk}$ | $AP^{mk}_{50}$ | $AP^{mk}_{75}$ | $AP^{bb}$ | $AP^{bb}_{50}$ | $AP^{bb}_{75}$ | $AP^{mk}$ | $AP^{mk}_{50}$ | $AP^{mk}_{75}$ |
| Supervised | 90 | 19.4 | 36.6 | 18.6 | 18.3 | 33.5 | 17.9 | 24.7 | 43.1 | 25.3 | 22.9 | 40.0 | 23.4 |
| SoCo | 100 | 24.6 | 42.2 | 25.6 | 22.1 | 38.8 | 22.4 | 29.2 | 47.7 | 31.1 | 26.1 | 44.4 | 27.0 |
| SoCo | 400 | 26.0 | 43.2 | 27.5 | 22.9 | 40.0 | 23.4 | 30.4 | 48.6 | 32.5 | 26.8 | 45.2 | 27.9 |
| **SoCo\*** | 400 | **26.8** | **45.0** | **28.3** | **23.8** | **41.4** | **24.2** | **31.1** | **49.9** | **33.4** | **27.6** | **46.4** | **28.9** |

Table 9: Pretraining on non-object-centric dataset.

| Pretraining dataset | Epoch | $AP^{bb}$ | $AP^{bb}_{50}$ | $AP^{bb}_{75}$ | $AP^{mk}$ | $AP^{mk}_{50}$ | $AP^{bb}_{75}$ |
|---|---|---|---|---|---|---|---|
| ImageNet | 100 | 42.3 | 62.5 | 46.5 | 37.6 | 59.1 | 40.5 |
| ImageNet-subset | 100 | 36.9 | 56.2 | 40.1 | 33.2 | 53.0 | 35.6 |
| COCO train set + unlabeled set | 100 | 37.3 | 56.5 | 40.6 | 33.5 | 53.4 | 36.0 |
| COCO train set + unlabeled set | 530 | 40.6 | 61.1 | 44.4 | 36.4 | 58.1 | 38.7 |

as the evaluation set. The other settings remain unchanged. Table 8 summarizes the results. Compared with the supervised pretraining baseline which achieves 19.4 $AP^{bb}$ / 18.3 $AP^{mk}$ and 24.7 $AP^{bb}$ / 22.9 $AP^{mk}$ on the 5% and 10% Mini COCO benchmarks, SoCo obtains large improvements of +6.6 $AP^{bb}$ / +4.6 $AP^{mk}$ and +5.7 $AP^{bb}$ / +3.9 $AP^{mk}$, respectively. SoCo\* further boosts performance with improvements of +7.4 $AP^{bb}$ / +5.5 $AP^{mk}$ and +6.4 $AP^{bb}$ / +4.7 $AP^{mk}$ over the supervised pretraining baseline.

**Pretraining on Non-object-centric Dataset.** We use COCO training set and unlabeled set as pretraining data. Since ImageNet is ∼5.3 times larger than COCO, we pretrain our method on COCO for 100 epochs and prolonged 530 epochs respectively. Furthermore, we also generate a dataset named ImageNet-Subset which contains the same number of images of COCO to verify the impact of different pretraining datasets. Table 9 shows the results. Compared with the ImageNet pretrained model, the COCO pretrained model under 530 epochs is lower by 1.7 AP, possibly because the scale of COCO is smaller than ImageNet. Compared with the ImageNet-Subset pretrained model, the COCO pretrained model under 100 epochs is +0.4 AP higher, which demonstrates the importance of domain alignment between the pretraining dataset and detection dataset.

**Longer Finetuning.** We transfer SoCo\* to Mask R-CNN with R50-FPN under the COCO 4× schedule. Table 10 shows the results. Our model continues to improve the performance with the 4× finetuning schedule, surpassing the supervised baseline by +2.6 $AP^{bb}$/ +1.9 $AP^{mk}$.

Table 10: Finetuning on **COCO** 4× schedule using Mask R-CNN with **R50-FPN**.

| Method | Epoch | $AP^{bb}$ | $AP^{bb}_{50}$ | $AP^{bb}_{75}$ | $AP^{mk}$ | $AP^{mk}_{50}$ | $AP^{bb}_{75}$ |
|---|---|---|---|---|---|---|---|
| Supervised | 90 | 41.9 | 61.5 | 45.4 | 37.7 | 58.8 | 40.5 |
| **SoCo\*** | 400 | **44.5** | **64.2** | **48.8** | **39.6** | **61.5** | **42.4** |

# 5 Conclusion

In this paper, we propose a novel object-level self-supervised pretraining method named Selective Object COntrastive learning (SoCo), which aims to align pretraining to object detection. Different from prior image-level contrastive learning methods which treat the whole image as an instance, SoCo treats each object proposal generated by the selective search algorithm as an independent instance, enabling SoCo to learn object-level visual representations. Further alignment is obtained in two ways. One is through network alignment between pretraining and downstream object detection, such that all layers of the detectors can be well-initialized. The other is by accounting for important properties of object detection such as object-level translation invariance and scale invariance. SoCo achieves state-of-the-art transfer performance on COCO detection using a Mask R-CNN detector. Experiments on two-stage and single-stage detectors demonstrate the generality and extensibility of SoCo.

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
