In the appendix, we present details of the selective search algorithm, the ImageNet linear evaluation results and the broader impact of this work.

# A  Selective Search

## A.1  Implementation Details

There are mainly three parameters in the selective search approach: $scale$, $\sigma$ and $min\_size$. The parameter $scale$ controls the number and size of the produced segments, that higher $scale$ means less but larger segments. The parameter $\sigma$ is the diameter of the Gaussian kernel used for smoothing the image prior for segmentation. The parameter $min\_size$ denotes the minimum component size. We use the default values of this approach: $scale = 500$, $\sigma = 0.9$ and $min\_size = 10$.

## A.2  Visualization

Figure 1 shows the proposals generated by the selective search approach, which shows reasonably good to cover the objects.

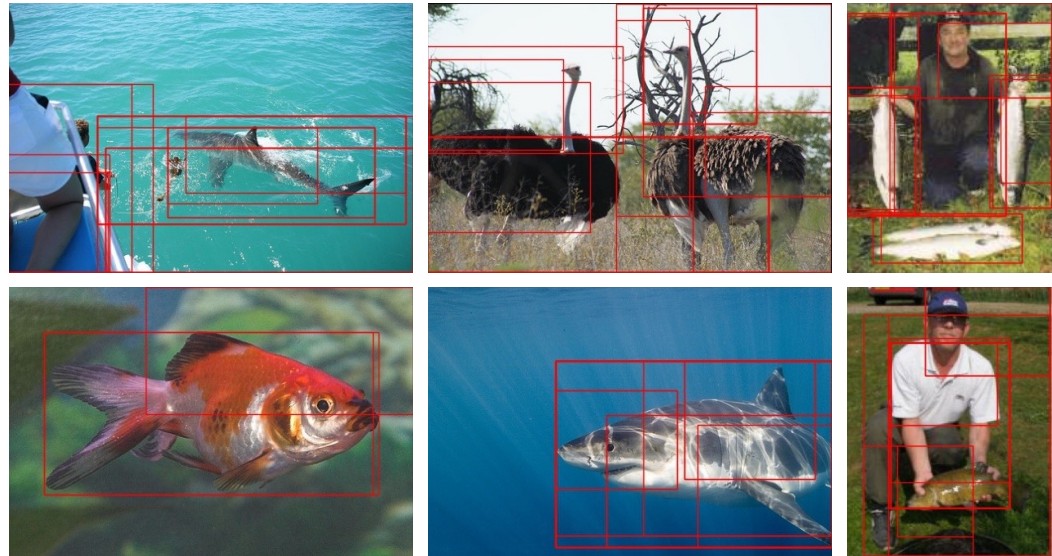

Figure 1: Proposals found by the selective search approach. The visualized proposals are randomly drawn from all proposals in each image for clear view.

## A.3  Statistics

The distribution of proposal number and size in each image are shown in Figure 2.

# B  Linear Evaluation on ImageNet-1K

In this section, we present the ImageNet-1K linear evaluation results for reference.

Only the backbone (ResNet-50) weights are leveraged for ImageNet linear classification, and all of the dedicated modules (e.g. FPN) are dropped. Our linear classifier training follows common practice [1, 5, 6, 2]: random crops with resize of $224 \times 224$ pixels, and random flips are used for data augmentation. The backbone network parameters and the batch statistics are both fixed. The classifier is trained for 100 epochs, using a SGD optimizer with a momentum of 0.9 and a batch size of 256. The initial learning rate is set to 30 and the weight decay is set to 0. In testing, each image is first resized to $256 \times 256$ pixels using bilinear resampling and then center cropped to $224 \times 224$.

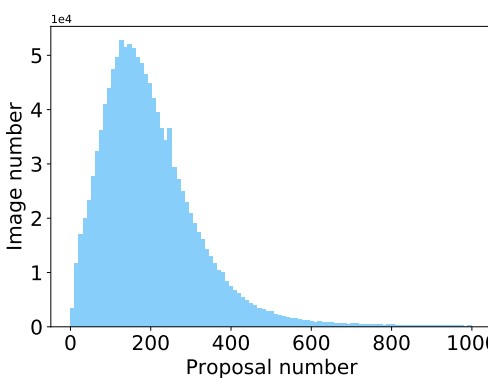 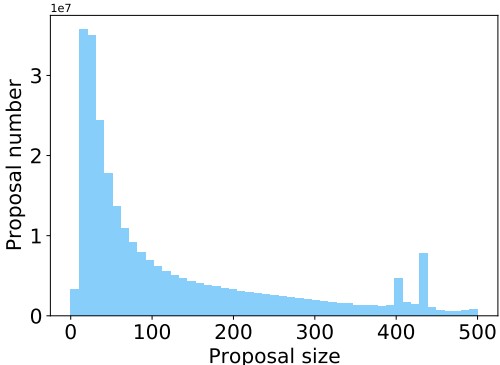

Figure 2: (Left) Histogram of the proposal number per image. (Right) Histogram of proposal size. We denote the proposal size by $\sqrt{hw}$, where $h$ and $w$ are the height and width of the proposal box, respectively.

Table 1: Comparison with state-of-the-art methods on ImageNet-1K linear evaluation with the **ResNet-50** backbone. The table is split to two sub-tables for better placement.

(a) Sub-table 1.

| Methods | Epoch | Top-1 | Top-5 |
|---------|-------|-------|-------|
| Supervised | 90 | 76.5 | - |
| SoCo (C4) | 100 | 59.7 | 82.8 |
| SoCo (C4) | 400 | 62.6 | 84.6 |
| SoCo (FPN) | 100 | 53.0 | 77.5 |
| SoCo (FPN) | 400 | 54.2 | 79.5 |
| SoCo* (FPN) | 400 | 53.9 | 79.2 |

(b) Sub-table 2.

| Methods | Epoch | Top-1 | Top-5 |
|---------|-------|-------|-------|
| MoCo [1] | 200 | 60.6 | - |
| SimCLR [2] | 1000 | 69.3 | 89.0 |
| MoCo v2 [3] | 800 | 71.1 | - |
| InfoMin [4] | 800 | 73.0 | 91.1 |
| BYOL [5] | 1000 | 74.3 | 91.6 |
| SwAV [6] | 800 | 75.3 | - |
| SimSiam [7] | 800 | 71.3 | - |

Table 1 reports top-1 and top-5 accuracies (%) on the ImageNet-1K validation set. The performance of SoCo is lower than previous image-level self-supervised pretraining methods. We expect joint image-level and object-level tasks could bridge this gap, and will be left as our future exploration.

## C    Data Efficiency

We conduct experiments to verify the data efficiency of our method. Concretely, we randomly select 50% of the training data from COCO dataset for SoCo* finetuning, the result is compared with the model trained on full COCO training data by using supervised pretraining. We use the standard COCO $1\times$ setting. Table 2 shows the comparison, our method has $2\times$ data efficiency compared with supervised pretraining.

Table 2: Data efficiency experiment by using Mask R-CNN with **R50-FPN**.

| Method | Data amount | $AP^{bb}$ | $AP^{bb}_{50}$ | $AP^{bb}_{75}$ | $AP^{mk}$ | $AP^{mk}_{50}$ | $AP^{bb}_{75}$ |
|--------|-------------|-----------|----------------|----------------|-----------|----------------|----------------|
| Supervised | 100% | 38.9 | 59.6 | 42.7 | 35.4 | 56.5 | **38.1** |
| **SoCo*** | 50% | **40.1** | **60.2** | **43.9** | **35.7** | **57.0** | 37.9 |

## D    Broader Impact

This work aims to design a self-supervised pretraining method for object detection. Any object detection applications may benefit from this work. There may be unpredictable failures. The consequences of failures by this algorithm are determined on the down-stream applications, and

please do not use it for scenarios where failures will lead to serious consequences. The method is data driven, and the performance may be affected by the biases in the data. So please also be careful about the data collection process when using it.