# OpenReview forum: "Aligning Pretraining for Detection via Object-Level Contrastive Learning"
_NeurIPS.cc/2021/Conference — NeurIPS 2021 Spotlight_

### Official Review · Reviewer_SV22 · 2021-07-05

**Rating:** 7
**Confidence:** 4

**Summary:**

This submission addresses the problem of constructing a good initialized pretrained network for the object detection task. Compared to previous work, the authors argue that pretraining should be adapted to object detection by focusing on object-level features learning. Instead of pretraining on ImageNet, the authors pretrain the model by designing an object level contrastive learning framework. Experiments on COCO and PASCAL show that the proposed method achieves higher performance.

**Limitations And Societal Impact:**

* The title may be too broad. It would be better to limit to two-stage detector, or the authors need to verify such method also works well on one-stage or anchor-free detectors.
* To show the proposed method is insensitive to the proposal generator's selection, it would be better to choose different proposal generator other than Selective Search and do parallel experiments
* The authors may need to address the imbalance of the proposals for different layers' learning problem as mentioned in the weakness part above.

**Main Review:**

### Strengths
+ The motivation and novelty of the submission is clearly explained. The Figure 1 is almost self-explainable for the framework.

+ The details of implementation are provided in the submission. Besides, the authors also provide reproducible code in the supplementary.

+ The authors provide ablation study for different aspects, including hyper-parameters, different datasets to verify the proposed method, which is much appreciated.

### Weakness
- Limited application scope: As claimed in the title, this proposed method is used for "Pretraining for Detection". This means this method should be generalized for all kinds of detectors for object detection. However, the major novelty is proposing a pretraining framework which focuses on object-level feature learning by using some unsupervised proposal generators. This method may perform well on two-stage detectors (also verified in the experiments). However, for other types of detectors, such as one-stage detectors and anchor-free detectors, there are no experiments to verify.

- Ablation studies of the proposal generator: There are two aspects which may be needed to make the method more convincing.
  * In this submission, the authors only adopt Selective Search which is an old unsupervised proposal generator. The influence of different selections of proposal generators is not verified. If the proposed method is not sensitive to such selection, the value of the submission can further increase.
  * According to Figure 2 in Section A.3 (Supplementary), the distribution of proposal size highly concentrates on the size of 20-60. However, in line 168-169, the authors assign different proposals to different layers of FPN according to their area. This may influence the learning effects of lateral layers ($P_3, P_4, P_5$), because #proposals is small.

**Time Spent Reviewing:**

4

---

> ### Author Response · Authors · 2021-08-10
> **Response to Reviewer SV22**
>
> Thanks for your constructive comments. We respond to them below.
>
> 1. **Limited application scope.**
>
>     In addition to two-stage detectors, we further conduct experiments on RetinaNet and FCOS, which are representative works for single-stage and anchor-free detectors. We transfer the weights of the backbone and FPN layers learned by our method to the RetinaNet and FCOS architectures. We follow the standard COCO 1$\times$ schedule and use the default optimization configs.
> The following table shows the comparison. The improvement over the supervised baseline demonstrates the effectiveness and generality of our method.
> We will include these results in the revision.
>
> 	| Detector  | Pretraining | AP$^{\text{bb}}$   | AP$^{\text{bb}}_{50}$ | AP$^{\text{bb}}_{75}$ |
> 	|:---------|:-----------|:-----------|:--------------|:--------------|
> 	| RetinaNet | Supervised  | 36.3        | 55.3           | 38.6           |
> 	| RetinaNet | SoCo*       | 38.3 (+2.0) | 57.2 (+1.9)    | 41.2 (+2.6)    |
> 	| FCOS      | Supervised  | 36.6        | 56.0           | 38.8           |
> 	| FCOS      | SoCo*       | 37.4 (+0.8) | 56.3 (+0.3)    | 39.9 (+1.1)    |
>
> 2. **Ablation studies of the proposal generator.**
>
>     As suggested, we use Edge Boxes [1] as another proposal generator to study the impact of different proposal generation methods. We use the same data pre-processing as in the Selective Search method. The following table shows the comparison with Selective Search and random proposal generation (which is discussed in L299-312 and Table 5c). We will add this ablation study in the revision.
>
> 	| Proposal generator | Proposal number | AP$^{\text{bb}}$ | AP$^{\text{bb}}_{50}$ | AP$^{\text{bb}}_{75}$ |
> 	|:------------------|:----------------|:---------|:--------------|:--------------|
> 	| Random             | 1                | 41.4      | 61.1           | 45.5           |
> 	| Random             | 4                | NaN       | NaN            | NaN            |
> 	| Selective Search   | 4                | 42.3      | 62.5           | 46.5           |
> 	| Edge Boxes         | 4                | 42.2      | 62.6           | 46.0           |
>
> 3. **The distribution of proposal size causes imbalanced learning.**
>
>     Figure 2 in Section A.3 shows the distribution of raw proposal size. Though the raw proposal size is highly concentrated at 20-60, the random resize and crop data augmentation changes the size of the proposals significantly. As a result, each proposal has a chance to be assigned to different levels of FPN during training. The learning of different FPN layers is balanced.
>
> 4. **The title may be too broad.**
>
>     We agree that comprehensive experiments on broad architectures and detectors are needed to claim generality.
> As suggested, we further verify the effectiveness of our method on typical single-stage detectors, i.e. RetinaNet and FCOS. Our paper aims to deliver the information that alignment between pretraining and object detection is critical for effective transfer. We will apply our method on more detectors in the revision.
>
> ## References
>
> [1] Edge boxes: Locating object proposals from edges. Zitnick, C. Lawrence, and Piotr Dollár.

---

> > ### Comment · Reviewer_SV22 · 2021-08-25
> > **Thanks for the responses**
> >
> > Thanks for the authors' responses. The results address my questions. Thus, I would raise my rating to clear accept. Thanks for the efforts in rebuttal!

---

### Official Review · Reviewer_uFr7 · 2021-07-16

**Rating:** 7
**Confidence:** 4

**Summary:**

The paper aims to design a self-supervised pre-training method for the downstream object detection task.  The authors propose to perform self-supervised object-level representation learning. Specifically, this method uses selective search to obtain object proposals in unlabeled images and extract the object-level features as in Mask R-CNN. Clear improvements have been achieved on COCO object detection and instance segmentation with Mask R-CNN.

**Limitations And Societal Impact:**

Yes, they are in the appendix.

**Main Review:**

## Strength
- The proposed object-level pre-training method is simple, straightforward, and motivated.
- The paper is well written and organized, making it easy to read and follow.
- The method achieves strong performance in COCO semi-supervised setting (5%/10% training data).

## Weakness
- The used representation method as in BYOL is not exactly contrastive learning.  The method name in the title and main paper is not appropriate.  In the BYOL paper, the authors distinguish their method from contrastive methods.
- The fine-tuning experiments are mainly conducted on COCO with a single architecture Mask R-CNN. But the pre-training should be general, even in the single task of object detection. To prevent overfitting a single dataset and a single arch. The authors are suggested to experiment with different datasets, e.g., Cityscapes, and different architectures, e.g., RetinaNet and FCOS. For different archs, you can just extract the pre-trained backbone and compare it to supervised or other self-supervised methods.
- The proposed method performs worse than the previous method PixPro on VOC.
- In Table 4, why does FPN make the performance worse?

## Others
- Did you try pre-training on non-object-centric datasets like COCO? The paper only reports results on ImageNet which is object-centric. I guess that's also why you can only set a small K for each image. It would be more interesting if the method can benefit more from data like COCO.

## After Rebuttal
The rebuttal has addressed all my concerns. Thus I raise the rating from 6 to 7.


**Time Spent Reviewing:**

6 hours

---

> ### Author Response · Authors · 2021-08-10
> **Response to Reviewer uFr7**
>
> Thanks for your constructive comments. We respond to them below.
>
> 1. **BYOL is not exactly contrastive learning.**
>
>     Thanks for pointing this out. We agree that BYOL is actually not a contrastive method since it does not rely on negative samples. We will consider changing the name of our method from "Selective Object Contrastive Learning" to "Selective Object Pretraining".
>
> 2. **Pretraining on non-object-centric dataset.**
>
>     We use ImageNet as the pretraining dataset for a fair comparison with existing methods. We strongly agree that we should use other large detection datasets such as OpenImages or Object365 for pretraining.
> Due to limited resources, we investigate this experiment on the combination of COCO training set and unlabeled set as suggested.
> Since ImageNet is ~5.3 times larger than COCO, we pretrain our method on COCO for 100 epochs and prolonged 530 epochs respectively. Furthermore, we also generate a dataset named ImageNet-Subset which contains same number of images of COCO to verify the impact of different pretraining datasets. The following table shows the results and the comparison with the models trained on ImageNet and ImageNet-Subset. Compared with the ImageNet pretrained model, the COCO pretrained model under 530 epochs is lower by $1.7$ $AP$, possibly because the scale of COCO is smaller than ImageNet. Compared with the ImageNet-Subset pretrained model, the COCO pretrained model under 100 epochs is +$0.4$ $AP$ higher, which demonstrates the importance of domain alignment between the pretraining dataset and detection dataset. Due to the rich contextual information in the object detection dataset, a larger number of object proposals may be applied for our model to improve the performance. We will continue to explore this direction in the future.
>
> 	| Pretraining dataset                  | Epoch | AP$^{\text{bb}}$ | AP$^{\text{bb}}_{50}$ | AP$^{\text{bb}}_{75}$ | AP$^{\text{mk}}$ | AP$^{\text{mk}}_{50}$ | AP$^{\text{mk}}_{75}$ |
> 	|:-----------------------------------|:-----|:--------|:--------------|:--------------|:---------|:--------------|:--------------|
> 	| ImageNet                            | 100   | 42.3     | 62.5           | 46.5           | 37.6      | 59.1           | 40.5           |
> 	| ImageNet-Subset                     | 100   | 36.9     | 56.2           | 40.1           | 33.2      | 53.0           | 35.6           |
> 	| COCO train set + COCO unlabeled set | 100   | 37.3     | 56.5           | 40.6           | 33.5      | 53.4           | 36.0           |
> 	| COCO train set + COCO unlabeled set | 530   | 40.6     | 61.1           | 44.4           | 36.4      | 58.1           | 38.7           |
>
> 3. **Transfer learning on different object detectors.**
>
>     Thanks for the suggestion. In addition to two-stage detectors, we further conduct experiments on RetinaNet and FCOS, which are representative works for single-stage and anchor-free detectors. We transfer the weights of backbone and FPN layers learned by our method to RetinaNet and FCOS architectures. We follow the standard COCO 1$\times$ schedule and use the default optimization configs. The following table shows the comparison. The improvement over the supervised baseline demonstrates the effectiveness and generality of our method. We will include these results in the revision.
>
> 	| Detector  | Pretraining | AP$^{\text{bb}}$   | AP$^{\text{bb}}_{50}$ | AP$^{\text{bb}}_{75}$ |
> 	|:---------|:-----------|:-----------|:--------------|:--------------|
> 	| RetinaNet | Supervised  | 36.3        | 55.3           | 38.6           |
> 	| RetinaNet | SoCo*       | 38.3 (+2.0) | 57.2 (+1.9)    | 41.2 (+2.6)    |
> 	| FCOS      | Supervised  | 36.6        | 56.0           | 38.8           |
> 	| FCOS      | SoCo*       | 37.4 (+0.8) | 56.3 (+0.3)    | 39.9 (+1.1)    |
>
> 4. **Transfer learning on extra dataset.**
>
>     In addition to COCO and PASCAL VOC object detection, we also consider the challenging LVIS v1 dataset to demonstrate the effectiveness and generality of our approach. The LVIS v1 detection benchmark contains 1203 categories in a long-tailed distribution with few training samples. It is considered to be more challenging than the COCO benchmark. We use Mask R-CNN with R50-FPN backbone and follow the standard LVIS 1$\times$ and 2$\times$ training schedules. We do not use any training or post-processing tricks.
> The following table shows the results. We can see that our method achieves +$5.9$ $AP^{\text{bb}}$ / +$5.6$ $AP^{\text{mk}}$ and +$4.9$ $AP^{\text{bb}}$ / +$4.6$ $AP^{\text{mk}}$ improvements under the LVIS 1$\times$ and 2$\times$ training schedules, respectively.
>
> 	| Pretraining | LVIS | AP$^{\text{bb}}$   | AP$^{\text{bb}}_{50}$ | AP$^{\text{bb}}_{75}$ | AP$^{\text{mk}}$   | AP$^{\text{mk}}_{50}$ | AP$^{\text{mk}}_{75}$ |
> 	|:-----------|:----|:-----------|:--------------|:--------------|:-----------|:--------------|:--------------|
> 	| Supervised  | 1x   | 20.4        | 32.9           | 21.7           | 19.4        | 30.6           | 20.5           |
> 	| SoCo*       | 1x   | 26.3 (+5.9) | 41.2 (+8.3)    | 27.8 (+6.1)    | 25.0 (5.6)  | 38.5 (+7.9)    | 26.8 (+5.7)    |
> 	| Supervised  | 2x   | 23.4        | 36.9           | 24.9           | 22.3        | 34.7           | 23.5           |
> 	| SoCo*       | 2x   | 28.3 (+4.9) | 43.5 (+6.6)    | 30.7 (+5.8)    | 26.9 (+4.6) | 41.1 (+6.4)    | 28.7 (+5.2)    |
>
> 5.  **The performance is worse than PixPro on VOC dataset.**
>
>     VOC is dominated by large sized objects, while objects in COCO are balanced across scales.
> Our method proposes scale-invariant object-level representations and thus achieves balanced performance across various scales. The following tables report: 1) $AP_\text{s}$, $AP_\text{m}$ and $AP_\text{l}$ of our method and PixPro on the COCO dataset; 2) object size distribution of the COCO and VOC datasets. Compared with PixPro, our method is advantageous especially for small objects.
>
> 	| Method | Dataset | AP      | AP$_{\text{s}}$ (area $<32^2$)  | AP$_{\text{m}}$ ($32^2 \leq$ area $<96^2$) | AP$_{\text{l}}$ (area $\geq 96^2$) |
> 	|:------|:-------|:--------|:-----------------------|:-----------------------------------|:---------------------------|
> 	| PixPro | COCO    | 41.4    |  24.8                    | 44.8                                | 54.3                        |
> 	| SoCo*  | COCO    | 43.2 (+1.8)  | 27.7 (+2.9)         | 46.4 (+1.6)                         | 56.0 (+1.7)                 |
>
> 	| Dataset | Small (%) | Medium (%) | Large (%) |
> 	|:-------|:---------|:----------|:---------|
> 	| COCO    | 41        | 34         | 25        |
> 	| VOC     | 9         | 31         | 60        |
>
> 6. **FPN makes the performance worse in Table 4.**
>
>     We discuss this phenomenon in L282-287. Only introducing FPN into pretraining slightly hurts the performance while including both FPN and the R-CNN head improves the transfer performance. We suspect that this minor decrease is due to sub-optimal optimization for FPN layers during finetuning.

---

> > ### Comment · Reviewer_uFr7 · 2021-08-25
> > **Thanks for the rebuttal**
> >
> > Thank you for the informative rebuttal. The responses have addressed all my concerns.
> > The added experiments make the paper much stronger, especially the pre-training over different datasets, the fine-tuning on different detectors and datasets.
> > I'm happy to raise my rating to a clear accept.;)

---

### Official Review · Reviewer_SXPo · 2021-07-16

**Rating:** 8
**Confidence:** 4

**Summary:**

This paper proposes SoCo, a method for self-supervised representation learning for downstream object detection and instance segmentation tasks.

They argue for stronger alignment between downstream tasks and self-supervised pre-training. They demonstrate this in the case of object detection by (a) developing a representation learning scheme that encourages object level scale and translation invariance and (b) pre-training several components of the downstream detector such as FPN and R-CNN head unlike prior work which mostly just trains the ResNet backbone.

The former property will help their model detect objects at several locations and scales as is needed for object detection.  The latter property helps them get stronger alignment between the upstream and downstream setups. To this end they propose "scale aware assignment". The ideas is that, representations for small objects are taken from finer FPN pyramid levels and for bigger objects are taken from coarser FPN pyramid levels. This matches what happens in the actual FPN where anchors at finer pyramid levels are smaller and at coarser pyramid levels are bigger.

The overall recipe is as follows: Selective search, a hand-crafted region proposal technique, is used to suggest boxes on all images. K of these boxes are randomly selected in each training step per image and three views of that image are constructed using random resized crop and resize operations. FPN features are extracted using RoiAlign for these box locations using "scale aware assignment". A BYOL loss is applied between representations of the same box across the three views. As in BYOL an EMA teacher network and projection head are used.

They pretrain R50-C4 and R50-FPN  models upstream and fine-tune corresponding Mask-RCNN models downstream for MS COCO and Pascal VOC object detection and MS COCO instance segmentation. Low data regime is explored using 5% and 10% "Mini" COCO setups. ImageNet linear eval results are included in the appendix. Ablation experiments evaluate whether stronger alignment in terms of architecture between pre-training and fine-tuning were helpful and sensitivity w.r.t. hyper-parameters such as view size, momentum for EMA, batch-size etc.

**Limitations And Societal Impact:**

The model does poorly on ImageNet classification downstream. This limitation has been identified and reported in the appendix. Societal impact has also been discussed.

**Main Review:**

The paper is well written and easy to understand. The experiments section is detailed using various datasets and different amounts of data and meaningful ablation experiments. Self-supervised learning for detection is of significance to the community and this paper furthers an emerging theme of trying to pre-train the entire model and do more than just learn a global image representation.

Recommendations to further improve the paper are below.

InsLoc [12], DetCon [13], and ReSim [41] are all closely related to the submission. Of these only InsLoc is explained in some detail in the related works section. DetCon suggests regions and aligns representations across multiple views. ReSim uses a sliding window approach instead of heuristic region proposal methods, but in the end also identifies positive pairs based on spatial alignment and uses a contrastive loss to align representations. $\mathrm{ReSim-FPN}^T$ infact transfers the FPN downstream similar to how the proposed method does. But ReSim does not have "scale aware assignment". ReSim and DetCon are contemporary works to this submission. They came out on arxiv just 2 months before the deadline, but a more detailed discussion of these three papers will be a great addition to the related works section.

Please also add "Self-Supervised Representation Learning from Flow Equivariance. Yuwen Xiong, Mengye Ren, Wenyuan Zeng, Raquel Urtasun Tech report, arXiv, Jan. 2021" (This is a dense representation learning method that uses flow as an additional data augmentation).

Also "UP-DETR: Unsupervised Pre-training for Object Detection with Transformers. Zhigang Dai, Bolun Cai, Yugeng Lin, Junying Chen. CVPR 2021" is similar to the current work as it pre-trains all of DETR, and not just the backbone, and transfers it downstream.

Lastly Romijnders, Rob, et al. "Representation learning from videos in-the-wild: An object-centric approach." Proceedings of the IEEE/CVF Winter Conference on Applications of Computer Vision. 2021, has similar ideas but use a supervised object detector as their region proposal module and use predicted class labels to define positives.

In Table 5, is row 1 basically equal to BYOL? Is row-2 basically InsLoc? If not what are the differences?

In Table 5, how do you pre-train a FPN model without scale aware assignment in rows 3 and 4?

The performance deltas due to parameter changes in the ablation experiments are very small. This,  together with the lack of error bars, makes it difficult to know when a parameter is critical and when it has no effect. Computational costs might prohibit the addition of error bars everywhere but even showing just 1 number with error bars could convey a sense of the variances involved.

BYOL is not really contrastive. So calling it contrastive learning when there are no negatives involved is confusing. But I do not know what else to suggest.


## Post rebuttal update
The rebuttal response addresses my concerns. The rebuttal also included several experiments that further strengthen the paper. I continue to support the acceptance of this paper.

**Time Spent Reviewing:**

6

---

> ### Author Response · Authors · 2021-08-10
> **Response to Reviewer SXPo**
>
> Thanks for your constructive comments. We respond to them below.
>
> 1. **Related works.**
>
>     Thanks for pointing out the missing references. We will give more discussion to DetCon [1], ReSim [2] and cite the works [3-5] in the next revision.
>
> 2. **Settings in Table 5.**
>
>     **Row 1.** In Table 5, row 1 is basically BYOL under our hyper-parameter settings. We use the best hyper-parameters of our method to ablate the effectiveness for each component in Table 5.
>
>     **Row 2.** Row 2 is not equal to InsLoc. There are two main differences between row-2 and InsLoc: 1) as described in L77-81, InsLoc creates non-realistic images by copy-pasting one image onto another as pseudo object. Instead, we use the Selective Search algorithm to generate proposals as illustrated in Figure 1; 2) Our method has multiple object proposals in a real image while InsLoc only has one pseudo object in a synthetic image.
>
>     **Row 3 and row 4.** In row 3 and row 4, we assign each object proposal to all levels of FPN when scale aware assignment is not enabled. The loss for each object proposal is the average of losses across all levels of FPN.
>
>     We will clarify these in the revised manuscript.
>
> 3. **Error bars.**
>
>     It is a good suggestion to give error bars for all experiments. However, pretraining consumes too much resources as you noticed. Here, we train our model for four independent runs under 100 epochs and 400 epochs in the setting of Table 1. The following table shows that our method is quite robust across training runs.
>
> 	| Experiment ID | Epoch | AP$^{\text{bb}}$ | AP$^{\text{bb}}_{50}$ | AP$^{\text{bb}}_{75}$ | AP$^{\text{mk}}$ | AP$^{\text{mk}}_{50}$ | AP$^{\text{mk}}_{75}$ |
> 	|:-------------|:-----|:---------|:--------------|:--------------|:---------|:--------------|:--------------|
> 	| No.1          | 100   | 42.31     | 62.52          | 46.47          | 37.60     | 59.13          | 40.53          |
> 	| No.2          | 100   | 42.27     | 62.52          | 46.31          | 37.57     | 59.27          | 39.99          |
> 	| No.3          | 100   | 42.23     | 62.50          | 46.23          | 37.58     | 59.27          | 40.41          |
> 	| No.4          | 100   | 42.12     | 62.08          | 46.39          | 37.45     | 58.96          | 40.33          |
> 	| No.1          | 400   | 43.00     | 63.30          | 47.13          | 38.18     | 60.24          | 40.95          |
> 	| No.2          | 400   | 42.99     | 63.15          | 47.23          | 38.11     | 60.02          | 40.78          |
> 	| No.3          | 400   | 42.98     | 63.24          | 46.93          | 38.25     | 60.22          | 40.86          |
> 	| No.4          | 400   | 42.84     | 63.09          | 46.86          | 38.17     | 60.07          | 40.87          |
>
> 4. **BYOL is not contrastive.**
>
>     Thanks for pointing this out. We agree that BYOL is actually not a contrastive method since it does not rely on negative samples. We will consider changing the name of our method from "Selective Object Contrastive Learning" to "Selective Object Pretraining".
>
> ## References
>
> [1] Efficient visual pretraining with contrastive detection. Hénaff, Olivier J., et al.
>
> [2] Region similarity representation learning. Xiao, Tete, et al.
>
> [3] Self-Supervised Representation Learning from Flow Equivariance. Xiong, Yuwen, et al.
>
> [4] Up-detr: Unsupervised pre-training for object detection with transformers. Dai, Zhigang, et al.
>
> [5] Representation learning from videos in-the-wild: An object-centric approach. Romijnders, Rob, et al.

---

### Official Review · Reviewer_ABcQ · 2021-07-16

**Rating:** 9
**Confidence:** 5

**Summary:**

This paper provides a self supervised method tailored for object detection (in the RCNN
paradigm), inspired from BYOL.
The general idea is to use a non-parametric method (selective search) to generate some
boxes, then apply a BYOL-style loss between ROI-Aligned features of each box coming from
different augmentations of the image.

It differs from other SSL methods (BYOL, SWAV,...) in that it instead of pretraining solely
the backbone, this methods ensures that all the components of the detector (FPN, RCNN head)
are pre-trained.

It shows convincing results on COCO and pascal as well as some subsets, in a few shot setting.
The code is made available as part of the supplementary, thus helping with reproducibility.


**Limitations And Societal Impact:**

No particular societal impact.

**Main Review:**

Overall, the paper is very well written, clearly motivated and ablated. The contribution is
solid and seems widely applicable. The paper is lacking a few details and references that
I'm listing in the following section, but overall I would recommend acceptance.

## Missing references
Here are a couple of references which would make the related work more complete:

- [3] is a CVPR 2021 paper investigating SSL for detection in the DETR framework
- [4] explores SSL for object detection without using Imagenet images
- [5] has some similar contrastive ideas, targets the RetinaNet architecture
- [6,7] more recent SSL methods (for classification)

## Technical details on the SSL recipe

This paper is inspired by BYOL, but seem to diverge in a number of subtle way that are not
always clearly explained or ablated. I list them in this section.

### Stop gradients

Regular BYOL prevents gradients from flowing back to the target network, by using a
stop-gradient mechanism.
This is an important aspect of BYOL, it has been shown to be a mandatory for other methods eg [2].

In this paper, I didn't see any mention of a stop-gradient mechanism. Is it applied? If not
is there a reason for it? This needs to be clarified in my opinion.

### Symmetric loss

In section 3.3, the paper indicates that the contrastive loss is symmetrized by swapping V1
and V2/V3 and re-computing the loss.

This is not a cheap choice, since it incurs doubling the number of forwards. However, the
paper doesn't clearly motivate it, and as far as I can tell there is no ablation on the
impact this symmetrization has on the performance. Could this be clarified?

### Multi view

This paper uses 3 views in the loss, compared to BYOL which uses only 2.
This change is ablated in table 4, and improves performance by roughly 0.6 Box AP, while
having very little effect on the mask AP.

According to the paper, "multiple views (V3) further directs SoCo towards learning
scale-invariant and translation-invariant representations". Is there any evidence of that
fact? For example, if it really improves scale invariance, I would expect the performance on
small objects to improve more significantly than large, since the third view is a
down-sampled version of the image. Is this the case in practice? Similarly, table 5(a) would
possibly benefit from reporting the AP break-down by size.

What is the computational cost incurred by this third view?

In table 1, the paper further reports results with 4 views, with even better results.
It seems that adding views help, is there a reason to believe that it will saturate if one
kept adding more scales or crops?


Finally, the V3 is a deterministic transform of V2. Would it help to add some
stochasticity in the process? Eg resize to a random size (possibly bigger than the original)
instead of always downsampling by 2?


### Batchsize

The study on the batch size (table 5b) is interesting and a bit counter-intuitive. Usually
SSL methods tend to always benefit from bigger batch-size. Is there any intuition why this
is not the case for this particular case?

## Longer training?

In [1], the authors have shown that after sufficient iterations, the effect of imagenet
pre-training tends to be erased, and random init tends to perform as well supervised
pre-training.

This paper reports only short schedules (1x and 2x), which arguably are important for
practitioners. However, I would be interested to find out if this SoCo method keeps its edge
over the supervised baseline on a long schedule (say 6x)?

If it does, I would consider it an even more impressive result.

## Pretraining dataset

The method uses pre-training on ImageNet. While this aids comparison with various SSL
methods that are not detection aware, I would argue that this paper could be made stronger
by also using more complex images during pre-training. One potential option would be to
pre-train on the whole train set of COCO as well as the unlabeled set of COCO. If size is an
issue, other detection datasets could be appended (Object365, OpenImages, ...)

Beyond the pure performance implications, it would be interesting to see if some of the
conclusions of the ablations change for such a pretraining. In particular Table 5.c shows
that adding more randomly selected boxes doesn't help, and the paper blames it on the fact
that "most of the images in the ImageNet dataset only contain a few objects". Pretraining on
a real detection dataset would alleviate this issue, and potentially unlock the full
potential of the method.

## Localization pretraining?

In detection, both classification and localization play a big role in the final performance
of the model. Because of the contrastive criterion that is being used, it seems that more
emphasis is put on the classification part than on the regression part.

Would it be possible to include some regression-aware objectives in the pre-training as
well?
One idea that comes to mind is similar to the box-jitter idea that is used in this paper:
it could be possible to train the localization part of Mask-RCNN to cancel the jitter, ie
regress the original position of the box. Has anything along those lines been tried?


[1] Rethinking ImageNet Pre-training, He at al.

[2] Exploring Simple Siamese Representation Learning, Chen et al

[3] UP-DETR: Unsupervised Pre-training for Object Detection with Transformers, Dai et al

[4] Self-EMD: Self-Supervised Object Detection without ImageNet, Liu et al

[5] Unsupervised Pretraining for Object Detection by Patch Reidentification, Ding et al

[6] Barlow Twins: Self-Supervised Learning via Redundancy Reduction, J.Zbontar et al.

[7] Emerging Properties in Self-Supervised Vision Transformers, M.Caron et al.

# Post rebuttal update:
In light of the excellent rebuttal, I increase my score to 9

**Time Spent Reviewing:**

6

---

> ### Author Response · Authors · 2021-08-10
> **Response to Reviewer ABcQ**
>
> Thanks for your constructive comments. Our responses to them are given below.
>
> ## Missing references
> We will cite and discuss the related works [2-7] you mentioned in our next revision.
>
> ## Technical details on the SSL recipe
> 1. **Stop gradients.**
>
>     Following BYOL, we use an online network and a target network for optimization. As stated in L146-L149, the target network is the exponential moving average (EMA) of the online network. Thus there is a stop-gradient mechanism in the target network. Only the online network is back-propagated with gradients. We will clarify this in the paper.
>
> 2. **Symmetric loss.**
>
>     The symmetric loss is a widely used technique to stabilize convergence in self-supervised learning, such as in BYOL, SimSiam, and SimCLR. Our method also follows this common practice. As suggested, we conducted experiments to ablate the symmetric loss as shown in the following table. The symmetric loss formulation stabilizes learning and improves performance.
>
>    | Method                  | Epoch | AP$^{\text{bb}}$ | AP$^{\text{bb}}_{50}$ | AP$^{\text{bb}}_{75}$ | AP$^{\text{mk}}$ | AP$^{\text{mk}}_{50}$ | AP$^{\text{mk}}_{75}$ |
> 	|:-----------------------|:-----|:---------|:--------------|:--------------|:---------|:--------------|:--------------|
> 	|  SoCo w/ symmetric loss | 100   | 42.3      | 62.5           | 46.5           | 37.6      | 59.1           | 40.5           |
> 	| SoCo w/o symmetric loss | 100   | 35.9      | 54.8           | 39.0           | 32.4      | 51.8           | 34.6           |
>
>
> 3. **Multi view.**
>
>     The pretraining objective encourages consistency across the $V1$, $V2$ and $V3$ representations. In the following table, we break down the $AP$ into $AP_{\text{s}}$, $AP_{\text{m}}$ and $AP_{\text{l}}$. We find that introducing $V3$ actually promotes performance on medium and large sized objects, while hurting the detection of small objects. This is possibly because the $V3$ representation helps the learning of $V2$ and $V3$, but not vice versa. Using multi view training increases computation by ~30%.
>
> 	| Multi view    | Size of $V3$ | Epoch | AP       | AP$_{\text{s}}$ | AP$_{\text{m}}$ | AP$_{\text{l}}$ |
> 	|:-------------|:--------------|:-----|:--------|:--------|:--------|:--------|
> 	|               | -              | 100   | 41.7     | **27.3** | 45.1     | 53.5     |
> 	| $\checkmark$  | 96             | 100   | 42.1     | 25.9     | 45.3     | 54.2     |
> 	| $\checkmark$  | 112 (default)  | 100   | **42.3** | 26.4     | **45.8** | **55.3** |
> 	| $\checkmark$  | 128            | 100   | 42.1.    | 25.9     | 45.5     | 54.2     |
> 	| $\checkmark$  | 160            | 100   | 42.0     | 25.6     | 45.2     | 54.9     |
> 	| $\checkmark$  | 192            | 100   | 42.2     | 26.3     | 45.3     | 54.6     |
>
> 4. **More views.**
>
>     In Table 1, our method with four views achieves better performance. Due to limited resources during the rebuttal period, we were unable to conduct experiments with more views.
>
> 5. **Dynamic input.**
>
>     It is a good suggestion to use randomly resized images for pretraining since it introduces more data-driven variations for our method to learn better scale-invariant and translation-invariant representations.
> Concretely, we resize $V1$ and $V2$ to the size of $80-320$, and $V3$ to the size of $40-160$. We compare it with our default model in the following table and observe an improvement of +$0.2$ $AP^{\text{bb}}$ and +$0.2$ $AP^{\text{mk}}$. We will add this experiment in the revised paper.
>
> 	| Method              | Epoch | AP$^{\text{bb}}$   | AP$^{\text{bb}}_{50}$ | AP$^{\text{bb}}_{75}$ | AP$^{\text{mk}}$  | AP$^{\text{mk}}_{50}$ | AP$^{\text{mk}}_{75}$ |
> 	|:-------------------|:-----|:-----------|:--------------|:--------------|:----------|:--------------|:--------------|
> 	| SoCo (fixed size)   | 100   | 42.3        | 62.5           | 46.5           | 37.6       | 59.1           | 40.5           |
> 	| SoCo (dynamic size) | 100   | 42.5 (+0.2) | 62.8 (+0.3)    | 46.5 (+0.0)    | 37.8(+0.2) | 59.5(+0.4)     | 40.4(-0.1)     |
>
> 6. **Batch size.**
>
>     Table 5b shows that a batch size of $2048$ achieves the best performance. Increasing the batch size to $4096$ degrades the performance. Since our method selects $K$ object proposals per image, the actual object-level batch size amounts to $2048\times K$. As also observed by SimCLR [8], an overly large batch size may incur degraded performance. We will clarify this in the revision.
>
> ## Longer training
>
> We transfer our best model to Mask R-CNN with R50-FPN under the COCO 4$\times$ schedule and compare the results with models trained under the COCO 1$\times$ and 2$\times$ schedules. The following table shows the results.
> Our model continues to improve the performance with the 4$\times$ finetuning schedule, surpassing the supervised baseline by +$2.6$ $AP$. Due to limited resources during the rebuttal period, we were unable to examine even longer finetuning schedules.
>
>
> | Pretraining | COCO schedule |  AP$^{\text{bb}}$  |  AP$^{\text{mk}}$ |
> |:----------------|:-----------------|:--------------------------|:---------------------|
> | Supervised  | 4$\times$          | 41.9            | 37.7           |
> | SoCo*          | 4$\times$            | 44.5 (+2.6) | 39.6 (+1.9) |
> | Supervised  | 2$\times$            | 41.3            | 37.3           |
> | SoCo*          | 2$\times$            | 44.3 (+3.0) | 39.6 (+2.3) |
> | Supervised  | 1$\times$           | 38.9            | 35.4           |
> | SoCo*          | 1$\times$           | 43.2 (+4.3) | 38.4 (+3.0) |
>
> ## Pretraining dataset
>
> We use ImageNet as the pretraining dataset for a fair comparison with existing methods. We strongly agree that we should use other large detection datasets such as OpenImages or Object365 for pretraining.
> Due to limited resources, we investigate this experiment on the combination of COCO training set and unlabeled set as suggested.
> Since ImageNet is ~5.3 times larger than COCO, we pretrain our method on COCO for 100 epochs and prolonged 530 epochs respectively. Furthermore, we also generate a dataset named ImageNet-Subset which contains the same number of images of COCO to verify the impact of different pretraining datasets. The following table shows the results and the comparison with the models trained on ImageNet and ImageNet-Subset. Compared with the ImageNet pretrained model, the COCO pretrained model under 530 epochs is lower by $1.7$ $AP$, possibly because the scale of COCO is smaller than ImageNet. Compared with the ImageNet-Subset pretrained model, the COCO pretrained model under 100 epochs is +$0.4$ $AP$ higher, which demonstrates the importance of domain alignment between the pretraining dataset and detection dataset. Due to the rich contextual information in the object detection dataset, a larger number of object proposals may be applied for our model to improve performance. We will continue to explore this direction in the future.
>
> | Pretraining dataset                 | Epoch | AP$^{\text{bb}}$ | AP$^{\text{bb}}_{50}$ | AP$^{\text{bb}}_{75}$ | AP$^{\text{mk}}$ | AP$^{\text{mk}}_{50}$ | AP$^{\text{mk}}_{75}$ |
> |:-----------------------------------|:-----|:--------|:--------------|:--------------|:---------|:--------------|:--------------|
> | ImageNet                            | 100   | 42.3     | 62.5           | 46.5           | 37.6      | 59.1           | 40.5           |
> | ImageNet-Subset                     | 100   | 36.9     | 56.2           | 40.1           | 33.2      | 53.0           | 35.6           |
> | COCO train set + COCO unlabeled set | 100   | 37.3     | 56.5           | 40.6           | 33.5      | 53.4           | 36.0           |
> | COCO train set + COCO unlabeled set | 530   | 40.6     | 61.1           | 44.4           | 36.4      | 58.1           | 38.7           |
>
> ## Localization pretraining
>
> This is a very good suggestion!
> Classification and localization are two key ingredients for object detection. We believe that introducing localization pretraining will further align the pretraining to the downstream task, and hence make the results even more compelling. Our framework that utilizes object proposals naturally supports pretraining RPN and the box regression head. However, engineering the new system and tuning it is highly non-trivial. We leave it for future work.
>
> ## References
> [1] Rethinking ImageNet Pre-training, He at al.
>
> [2] Exploring Simple Siamese Representation Learning, Chen et al.
>
> [3] UP-DETR: Unsupervised Pre-training for Object Detection with Transformers, Dai et al.
>
> [4] Self-EMD: Self-Supervised Object Detection without ImageNet, Liu et al.
>
> [5] Unsupervised Pretraining for Object Detection by Patch Reidentification, Ding et al.
>
> [6] Barlow Twins: Self-Supervised Learning via Redundancy Reduction, J.Zbontar et al.
>
> [7] Emerging Properties in Self-Supervised Vision Transformers, M.Caron et al.
>
> [8] A simple framework for contrastive learning of visual representations, Chen et al.

---

> > ### Comment · Reviewer_ABcQ · 2021-08-21
> > **Thank you for the rebuttal**
> >
> > I thank the authors for the excellent rebuttal. I look forward to seeing the clarifications and additional experiments in the revised paper (and possibly appendix), as they completely cleared my doubts and give interesting insights on the method.
> >
> > The rebuttal has answered all my concerns, as well as many points raised by other reviewers.
> > In particular, I’m interested by the following observations:
> > - Despite being presumably tuned for Mask R-CNN, the rebuttal shows that the method is also applicable with good performance to one stage methods
> > - The performance on the LVIS dataset is very solid.
> > - The method keeps an edge over the supervised baseline on a 4x schedule, which I find significant.
> > - The method is promising on object centric dataset (eg coco) when controlling for dataset size. This is a very promising avenue, I’m hoping future work will explore this further (eg bigger object centric dataset, or combining Coco and IN,...)
> >
> > In light of this, I will raise my score.

---

### Decision · Program_Chairs · 2021-09-27

**Decision:**

Accept (Spotlight)

**Comment:**

All reviewers agree this is a solid paper with a good novel contribution, great experimental results, and detailed analysis/ablation. The authors did a good job addressing the reviewers' concerns in their responses too. Clear accept.